# Differential Effects of Insulin and IGF1 Receptors on ERK and AKT Subcellular Distribution in Breast Cancer Cells

**DOI:** 10.3390/cells8121499

**Published:** 2019-11-23

**Authors:** Rive Sarfstein, Karthik Nagaraj, Derek LeRoith, Haim Werner

**Affiliations:** 1Department of Human Molecular Genetics and Biochemistry, Sackler School of Medicine, Tel Aviv University, Tel Aviv 69978, Israel; rives@tauex.tau.ac.il (R.S.); mailkartz@gmail.com (K.N.); 2Division of Endocrinology, Diabetes and Bone Diseases, Department of Medicine, Icahn School of Medicine at Mount Sinai, New York, NY 10029, USA; derek.leroith@mssm.edu; 3Yoran Institute for Human Genome Research, Tel Aviv University, Tel Aviv 69978, Israel

**Keywords:** insulin-like growth factor-1 (IGF1), IGF1 receptor, insulin receptor, signaling

## Abstract

Insulin and insulin-like growth factor-1 (IGF1) have important roles in breast cancer development. The recent identification of nuclear insulin (INSR) and IGF1 (IGF1R) receptors provides a novel paradigm in the area of signal transduction. The fact that INSR and IGF1R can function as transcription factors, capable of binding DNA and controlling transcription, adds a new layer of biological complexity by conferring upon cell-surface receptors the ability to regulate genomic events. The present study was designed to assess the hypothesis that insulin and IGF1 pathways elicit differential effects on subcellular distribution and activation of ERK1/2 and AKT. To this end, MCF7 breast cancer-derived cell lines with specific INSR or IGF1R disruption were employed. In addition, small interfering RNA technology was used to specifically down-regulate INSR or IGF1R expression in T47D breast cancer cells. DNA affinity chromatography assays were conducted to address the specific binding of ERK1/2 and AKT to the *IGF1R* promoter region. We demonstrate that both INSR and IGF1R exhibit a nuclear localization in breast cancer-derived cells. In addition, the insulin and IGF1 pathways have different effects on the subcellular distribution (and, particularly, the nuclear presence) of ERK1/2 and AKT molecules. Both cytoplasmic mediators are capable of binding and transactivating the *IGF1R* promoter. In conclusion, our data are consistent with the notion that, in addition to their classical roles as targets for insulin-like molecules, both ERK1/2 and AKT are involved in transcriptional control of the *IGF1R* gene. This previously unrecognized regulatory loop may provide mechanistic advantages to breast cancer cells. Given the potential role of INSR and IGF1R as therapeutic targets in oncology, it will be of clinical relevance to address the future use of nuclear receptors and their downstream cytoplasmic mediators as biomarkers for INSR/IGF1R targeted therapy.

## 1. Introduction

Insulin and insulin-like growth factor-1 (IGF1) are regarded as important players in cancer biology [1,2]. A large body of clinical, experimental and epidemiological data supports the concept that activation of the insulin and IGF1 signaling pathways is a critical requisite for breast cancer progression [3,4,5]. Whereas the classical dogma maintains that insulin is responsible for metabolic types of action while IGF1 elicits primarily growth activities, there is a certain overlap in the spectrum of bioactivities of both hormones [6,7,8]. Furthermore, there is a large degree of cross-talk between the ligands and their cognate receptors, the insulin (INSR) and IGF1 (IGF1R) receptors. Thus, insulin at high concentrations can activate the IGF1R, leading to proliferative activities. On the other hand, IGF1 can stimulate the INSR, leading to metabolic types of action. The interplay between insulin-like hormones and their receptors is an area of major basic and clinical interest [9]. It is still unknown, however, why INSR and IGF1R, despite sharing most of their downstream signaling molecules, exhibit, for the most part, distinct, well-defined functions.

Recently, evidence has been provided indicating that both INSR and IGF1R can migrate to the nucleus in a wide range of normal and malignant cells [10,11,12,13]. While the mechanism responsible for nuclear import is yet to be discovered, nuclear INSR and IGF1R display an extended range of activities typically linked to transcription factors, including DNA binding and transcription regulation [14,15]. Of interest, nuclear IGF1R was shown to transactivate the *IGF1R* gene promoter, pointing to a novel mechanism of *IGF1R* positive autoregulation [12]. The ability of nuclear INSR and IGF1R to bind DNA in a sequence-specific fashion and to regulate transcription of genes involved in apoptosis and cell cycle control suggests that nuclear translocation of tyrosine kinase receptors may confer upon cells the ability to regulate growth and other cellular events at the genomic level [16,17].

The association of the IGF1 system with breast cancer development has been firmly established. Conflicting results, however, arose from the use of different methodologies, distinct molecular subtypes, and genetic differences between populations and tumor heterogeneity [18]. The IGF1R has emerged in recent years as a promising therapeutic target in oncology [19,20,21]. Unfortunately, the inherent complexity of this hormonal system, including the formation of hybrid receptors, hampered progress in the development of efficient pharmacological modalities [9,22]. Biochemical and molecular dissection of the mechanisms of action of insulin and IGF1 in breast cancer will be of major translational impact.

In view of the important roles of the INSR and IGF1R signaling pathways in breast cancer, we investigated the subcellular distribution of both receptors, as well as that of members of the extracellular signal-regulated kinases (ERK) and protein kinase B/AKT (PKB/AKT) families, two prototypical networks of cytoplasmic molecules involved in insulin/IGF1 signaling. The present study aimed at evaluating the hypothesis that insulin and IGF1 pathways elicit differential effects on subcellular distribution and activation of ERK1/2 and AKT. To this end, MCF7 and T47D breast cancer cells with disrupted INSR or IGF1R were employed. Data indicate that: (1) IGF1R silencing led to a marked reduction in nuclear ERK and AKT expression in MCF7 cells; (2) IGF1R, but not INSR, silencing had a major effect on nuclear ERK activation in MCF7 cells; (3) both ERK1/2 and AKT proteins are capable of binding and stimulating *IGF1R* promoter activity; (4) cells with a disrupted IGF1R exhibited enhanced proliferation, consistent with the notion that INSR signaling drives a stronger growth response in breast cancer. The interplay between the INSR/IGF1R pathways and the ERK and AKT effectors and, in particular, the nuclear and genomic interactions inherent to these networks, merits further investigation.

## 2. Materials and Methods

### 2.1. MCF7 Stable shRNA IGF1R/INSR Cell Lines

GIPZ plasmids encoding the following microRNA-adapted short hairpin RNAs (shRNA): TGACTGTGAAATCTTCGGC (human IGF1R) and CTTACCAAGGCCTGTCTAA3 (human INSR), packed in high titer lentiviral particles, were purchased from Open Biosystems (Huntsville, AL, USA). These plasmids or a plasmid containing a non-coding shRNA sequence (control shRNA) were transfected into breast carcinoma-derived estrogen receptor-positive (ER+) MCF7 cells (American Type Culture Collection, Manassas, VA, USA). All three vectors contain a green fluorescent protein (GFP) marker and a puromycin resistance gene. Transfected MCF7 cells were maintained in DMEM supplemented with 10% fetal bovine serum (FBS), 100 units/mL penicillin, 100 μg/mL streptomycin, 5.6 mg/L amphotericin B, and 1μg/mL puromycin. MCF7-derived cell lines were provided by Dr. Ran Rostoker (Technion, Haifa, Israel) and denominated IGF1R-KD and INSR-KD (or controls). In selected experiments, cells were treated with IGF1 [50 ng/mL (PeproTech Ltd., Rocky Hill, NJ, USA)] or insulin [50 ng/mL (Biological Industries Ltd., Bet-Haemek, Israel). All experiments were carried out at least twice.

### 2.2. T47D IGF1R/INSR siRNA Silencing

The T47D cell line, an ER+ breast cancer-derived line, was also employed in this study [23]. Unlike the MCF7 cell line that expresses a wild-type *p53* gene, the T47D cell line includes a mutant *p53* [24]. Small interference RNA (siRNA) used were: human IGF1R SMARTpool (L-003012-00-0005), human INSR SMARTpool (L-003014-00-0005) and non-targeting (NT) pool (D-001810-10-05). siRNAs were purchased from Dharmacon Research Inc (Lafayette, CO, USA). T47D cells were transfected using INTERFERin (Polyplus Transfection Inc, Illkirch, France). Briefly, cells were seeded into 6-well plates the day before transfection using a dose of 10 nM of siRNA. IGF1R or INSR knockdown was detected after 72 h by immunoblotting analysis. The T47D cell line was provided by Dr. Ilan Tsarfaty (Tel Aviv University, Israel). MCF7 and T47D cells are molecularly classified as Luminal A (ER^+^/PR^+^/HER2^−^).

### 2.3. Protein Analysis and Immunoblotting

Cells were harvested and lysed in a buffer containing proteases and phosphatases inhibitors. For preparation of cytosolic and nuclear extracts, cells were washed with phosphate-buffered saline (PBS) and the pellet was resuspended in a low salt buffer and spun (the supernatant is the cytosolic fraction). Nuclei-containing pellets were resuspended in high salt buffer. Protein concentration was determined using the Bradford reagent. Samples were electrophoresed through 10 or 12% SDS–PAGE, followed by protein transfer to nitrocellulose membranes. Blots were blocked with 5% skim milk and incubated with antibodies against: phospho-IGF1R/INSR (#3024), IGF1R β-subunit (#3027), INSR β-subunit (#3025), phospho-AKT (#9271), AKT (#9272), phospho-ERK1/2 (#9106) and ERK1/2 (#9102). Antibodies were from Cell Signaling Technology. An antibody against lamin B was obtained from Abcam (Cambridge, UK) and anti-heat shock cognate (HSC70) (#Hsp73) was from Santa Cruz Biotechnology (Dallas, TX, USA). Blots were washed and incubated with the appropriate horseradish peroxidase-conjugated secondary antibody. Proteins were detected using the enhanced chemiluminiscence reaction (Westar Supernova, Cyanagen, Bologna, Italy). HSC70 was used as a loading control.

### 2.4. Real-Time Quantitative Polymerase Chain Reactions (RT-QPCR)

Total RNA was prepared from stable cell lines using Trizol (ThermoFisher Scientific, Waltham, MA, USA). Total RNA (2 µg) was reverse transcribed using the High Capacity cDNA Reverse Transcription Kit (Applied Biosystems, Grand Island, NY, USA). IGF1R and INSR mRNA levels were measured by RT-QPCR, using specific primers (Table 1). RT-QPCR was performed using Faststart Universal SYBER green Master (Roche) and StepOne real-time PCR (Applied Biosystems). Amplification was carried out after an incubation of 2 min at 50 °C and 10 min at 95 °C, followed by 40 cycles at 95 °C for 15 s, 1 min at 60 °C, and 30 s at 72 °C. The number of PCR cycles to reach the fluorescence threshold was the cycle threshold (Ct). Each cDNA sample was tested in triplicate and mean C_t_ values are reported. For each reaction, a “no template” sample was included as a negative control. Fold differences were calculated using the 2^ΔΔC^_t_ method [25]. Levels of actin mRNA were used to normalize the different genes’ mRNA values.

### 2.5. Transient Co-Transfections

For transient co-transfection experiments an *IGF1R* promoter-luciferase reporter construct including 476 bp of 5’-flanking and 640 bp of 5’-untranslated regions of the *IGF1R* gene [p(−476/+640) luciferase (LUC)] was employed [26]. An expression vector encoding ERK2 (pEGFP-ERK2) was obtained from Addgene (Cambridge, MA, USA). A myr-AKT1 (constitutively active) and myr HA-AKT1 K179M (kinase domain dead) expression vectors (in pLCNX) were provided by Dr. Eran Perlson (Tel Aviv University). A GFP-IGF1R expression plasmid was provided by Dr. Rosemary O’Connor (University of Cork, Ireland). An INSR isoform A expression vector was a gift from Dr. Antonino Belfiore (University of Catania, Italy). Cells were seeded in 6-well plates the day before transfection and transfected with 1 μg of the *IGF1R* promoter reporter. IGF1 or insulin was added during the last 24 h in starvation medium. In some experiments, cells were co-transfected with 1 μg of the *IGF1R* promoter reporter along with 1μg of the ERK2 expression plasmid (or empty pCDNA3), or 1 μg of AKT1, or 1 μg AKT1 K179M (or empty pLCNX) in the presence or absence of GFP-IGF1R (1 μg) or GFP-INSR-A (1 μg) using the Jet-PEI transfection reagent (Polyplus Transfection Inc). Cells were harvested 48 h after transfection, and luciferase activity was measured as described [27]. Promoter activities are expressed as luciferase values normalized to protein concentrations.

### 2.6. Co-Immunoprecipitation (Co-IP) Assays

Total lysates (500–1000 µg), cytoplasmic and nuclear extracts (100–200 µg) were diluted 1:2 with IP dilution buffer [1% Triton X-100, 150 mM NaCl, 20 mM Tris buffer (pH 7.5) containing proteases and phosphatases inhibitors], and immunoprecipitated by incubating overnight at 4 °C with anti-IGF1R β-subunit or anti-INSR β-subunit, anti-ERK1 (K23), -ERK2 (C14) or -AKT. All antibodies were from Santa Cruz. Protein A/G-agarose beads (SC-20003; Santa Cruz) were added and incubated for 2 h. Samples were then washed with PBS, mixed with sample buffer, boiled for 10 min at 95 °C, and electrophoresed through 10% SDS-PAGE. Finally, membranes were blotted with anti-ERK1/2, anti-AKT, anti-IGF1R β-subunit, anti-INSR β-subunit or anti-SUMO-1, as described above.

### 2.7. Proliferation Assays

MCF7-derived stable transfectants were plated in 96-well plates in triplicate (6 × 10^3^ cells/well). Similarly, T47D cells were seeded 72 h post-siRNA transfection in triplicate wells (1 × 10^4^ cells/well). After 24 h, the medium was replaced with starvation media (DMEM with 5% FBS) in the presence or absence of IGF1 or insulin at a final concentration of 50 ng/mL for an additional 72 h (MCF7) or 48 h (T47D). Cell viability was determined by an XTT cell proliferation assay (Biological Industries). After 1 h, the colorimetric reaction was measured using an ELISA reader at a wavelength of 450 nm and a reference absorbance of 630 nm in at least three independent assays. Cell viability was expressed as a percentage of optical density values obtained upon treatment, relative to controls.

### 2.8. Migration Assays

Cell migration was assessed using scratch assays. Control, IGF1R-KD and INSR-KD MCF7 cells were seeded onto 96-well Image Lock microplate (#4379, Essen BioScience, Ann Arbor, MI, USA) for 16 h (3.5 × 10^4^ cells/well). After incubation, a scratch was done using WoundMaker (Essen BioScience) designed for these specific plates. The cells were then treated with starvation media (with 3% FBS) in the presence or absence of IGF1 or insulin for 96 h. The plates were placed into an IncuCyte ZOOM system (Essen Bioscience Inc., Ann Arbor, MI, USA), with repeat scanning every 2 h. Analysis of the results was done using the IncuCyte ZOOM software. Distance of cell migration was measured at middle position on the screen using Microsoft Windows PowerPoint.

### 2.9. DNA Affinity Chromatography

For DNA affinity chromatography, a biotinylated 511-bp human proximal *IGF1R* promoter fragment (extending from nucleotides −458 to +53) was bound to streptavidin magnetic beads (DynabeadsR M-270 Streptavidin; Dynak Biotech ASA, Oslo, Norway) and incubated with nuclear extracts of MCF7-derived stable transfectants [28]. The specifically-bound ERK and AKT proteins were detected by Western blotting.

### 2.10. Statistical Analyses

The statistical significance of the differences between groups was assessed by Student’s *t* test (two samples, equal variance). Scanning densitometry analyses were evaluated using TINA imaging analysis software (http://biochemlabsolutions.com/GelQuantNET.html). Signal intensities of phosphor-proteins were normalized to the corresponding protein signals.

## 3. Results

### 3.1. Subcellular Analysis of IGF1R and INSR Expression in Breast Cancer-Derived Cell Lines with Disrupted IGF1R/INSR

To evaluate the specific impact of insulin and IGF1 signaling on the subcellular distribution and activation of INSR and IGF1R, MCF7 breast cancer-derived cells with disrupted INSR (INSR-KD) or IGF1R (IGF1R-KD) were employed. Cellular fractionation experiments followed by Western blots indicate that control MCF7 cells expressed both INSR and IGF1R in cytoplasmic (membrane-containing) and nuclear fractions (Figure 1A). As expected, cells with disrupted INSR or IGF1R did not express the silenced receptor in neither fraction (Figure 1A–C). Use of an antibody directed against both phospho-INSR and phospho-IGF1R allowed the detection of the phosphorylated receptors in all subcellular compartments (Figure 2A,B). Given the almost total silencing achieved by the shRNA procedure, we assume that the phosphorylated receptor visualized in each cell line corresponds to the opposite (i.e., not silenced) receptor (Figure 1). Enhanced IGF1R levels were detected in both nuclear and cytoplasmic fractions of INSR-KD cells, suggesting the existence of a compensatory mechanism responsible for increased IGF1R expression following INSR silencing. Lamin B was used as a nuclear marker in fractionation experiments. To address the general nature of these observations, the experiments were replicated in the T47D cell line, an additional ER^+^ breast cancer-derived cell line. To this end, INSR and IGF1R expression was abrogated using specific siRNAs, as described in Section 2. As described above for MCF7 cells, T47D cells with disrupted INSR or IGF1R did not express the abrogated receptor in neither cellular fraction (Figure 1D–F). Of interest, INSR-KD cells expressed low levels of the INSR precursor (~250-kDa). Statistical analyses of the changes in IGF1R/INSR expression in subcellular fractions of both MCF7 and T47D cells following specific receptor silencing are presented in Figure 1B,C,E,F.

### 3.2. Effect of IGF1R/INSR Silencing on ERK and AKT Expression and Activation

To assess the effects of insulin and IGF1 on the subcellular expression and activation of downstream signaling molecules, levels of phospho- and total-AKT and ERK1/2 were measured in INSR-KD, IGF1R-KD and control MCF7 cells. Major reductions in total-AKT and total-ERK1/2 were seen in the nuclear fraction of IGF1R-KD, in comparison to control, cells (Figure 2A). Reductions in phospho-AKT were seen in cytoplasmic fractions of both INSR-KD and IGF1R-KD cells (Figure 2A,C). Of interest, IGF1R silencing had a major impact on nuclear ERK1/2 activation. INSR silencing, on the other hand, had no effect on ERK1/2 activation (Figure 2A,D). These results indicate that IGF1 signaling has a prominent role in nuclear ERK1/2 activation. Of interest, co-IP assays using anti-ERK1/2 or anti-AKT, along with a Sumo-1 antibody, revealed that nuclear ERK1/2 and AKT proteins were sumoylated, suggesting that this posttranslational modification might be of importance in nuclear transport of cytoplasmic mediators (Figure 2I,J). Analysis of T47D cells transfected with siRNA against INSR or IGF1R revealed a different pattern of IGF1R/INSR phosphorylation than in MCF7 cells. Thus, use of an antibody against the phosphorylated forms of both IGF1R and INSR detected a strong signal at the level of the receptor precursors (~250-kDa) (Figure 2E). We assume that the phosphorylated precursor detected in each cell line corresponds to the opposite (not silenced) pro-receptor. Phosphorylation of the mature receptors were only detected after longer exposure times (data not shown), probably reflecting a precursor-specific type of phosphorylation in T47D cells. Similarly to MCF7 cells, a decrease in phospho-AKT was seen in the nuclear fraction of IGF1R-KD T47D cells (Figure 2E,G). However, no reduction in nuclear phospho-ERK1/2 was seen in these cells (Figure 2E,H).

### 3.3. RT-QPCR of IGF1R and INSR mRNA Levels

Next, we examined whether changes in protein levels of IGF1R and INSR in MCF7 cells were accompanied by corresponding changes in mRNA levels. As shown in Figure 3A, IGF1R mRNA levels were reduced by 60% in IGF1R-KD cells, whereas they were increased by two-fold in INSR-KD cells. These results indicate that: (1) the reduction in IGF1R protein levels seen in IGF1R-KD cells was associated with a concomitant decrease in mRNA values; and (2) the increase in IGF1R mRNA levels in INSR-KD cells provides evidence of a compensatory mechanism that might be responsible for enhanced IGF1R expression and action in cells with a disrupted INSR. On the other hand, levels of INSR mRNA were reduced by 70% in INSR-KD cells and increased by 40% in IGF1R-KD cells (Figure 3B).

### 3.4. Effect of IGF1R/INSR Silencing on IGF1R Promoter Activity

To examine the impact of IGF1R/INSR silencing on basal and IGF1-stimulated *IGF1R* promoter activity, luciferase assays were conducted on IGF1R-KD and INSR-KD cells. To this end, MCF7 cells were transfected with an *IGF1R* promoter-luciferase reporter construct [p(−476/+640)LUC], in the presence or absence of IGF1 during the last 24 h of the incubation period. Results of transient transfections assays indicate that basal *IGF1R* promoter activity was reduced by 27% in IGF1R-KD, in comparison to control, cells (Figure 4). Furthermore, IGF1 was unable to stimulate promoter activity in IGF1R-KD cells, unlike control cells, in which IGF1 enhanced promoter activity by 192%. On the other hand, IGF1 enhanced *IGF1R* promoter activity in INSR-KD cells by 243% in comparison to untreated cells. The fact that basal *IGF1R* promoter activity was largely reduced in INSR-KD cells may indicate that activation of the *IGF1R* promoter requires an intact insulin signaling pathway. On the other hand, an intact IGF1R pathway is necessary for IGF1 activation of the *IGF1R* promoter. These results are consistent with previous studies showing that IGF1 exerts a stimulatory effect on *IGF1R* promoter activity [12].

### 3.5. Effect of ERK on IGF1R Promoter Activity

Next, we explored the ability of ERK2 to stimulate *IGF1R* promoter activity. These experiments were aimed at evaluating the hypothesis that, in addition to its classical role as a target for IGF1R/INSR action, ERK2 is able to regulate *IGF1R* gene expression at the transcriptional level. To this aim, cotransfection experiments were conducted in IGF1R-KD, INSR-KD and control MCF7 cells using an *IGF1R* promoter-luciferase reporter, along with an ERK2 expression vector. Luciferase measurements indicate that ERK2 stimulated *IGF1R* promoter activity in all three cell lines, although its effect was markedly reduced in IGF1R-KD and INSR-KD cells (Figure 5). Furthermore, cotransfection of an IGF1R, but not an INSR, expression vector reduced the ability of ERK2 to stimulate *IGF1R* promoter activity. These results indicate that *IGF1R* promoter activity was dependent on complex interactions involving downstream mediator molecule ERK2 and the cognate IGF1R protein.

### 3.6. Analysis of Physical Interactions between IGF1R and ERK1/2

To investigate potential differential physical interactions between INSR/IGF1R and ERK1/2 or AKT, co-IP assays were conducted in MCF7 cells with disrupted IGF1R or INSR. Results of co-IP experiments suggest that IGF1R and ERK1/2 exhibited an apparently stronger interaction in INSR-KD than in IGF1R-KD cells (Figure 5, inset). These results seem to suggest that ERK1/2 binds IGF1R with a larger affinity than it binds INSR.

### 3.7. Effect of AKT on IGF1R Promoter Activity

Next, we evaluated the ability of AKT to stimulate *IGF1R* promoter activity. Previous studies, as well as the present report, have identified AKT in the nuclei of various cell types [29]. The ability of AKT to regulate transcription of the *IGF1R* gene, however, has not yet been investigated. Cotransfection of an AKT expression vector along with an *IGF1R* promoter–luciferase reporter led to a 20-fold increase in promoter activity in control cells (Figure 6). The capacity of AKT to enhance IGF1R promoter activity was largely reduced in IGF1R-KD and INSR-KD cells. Thus, in IGF1R-KD cells AKT enhanced promoter activity only by ~11-fold, whereas, in INSR-KD cells, it was stimulated by ~12.5-fold. In addition, the effect of AKT was abrogated when co-expression assays were conducted in the presence of excess IGF1R or INSR. These results suggest that nuclear AKT may compete with endogenous IGF1R/INSR in transcriptional regulation of the *IGF1R* promoter. These functional interactions were corroborated by co-IP assays, showing physical interactions between AKT and IGF1R/INSR (Figure 6, inset). Of interest, cotransfection of a mutant, kinase-dead AKT expression vector (AKT179M), along with the *IGF1R* promoter reporter, led to a highly enhanced stimulatory effect (~twice the effect of wild-type AKT) (Figure 7). These results indicate that AKT phosphorylation is not required for *IGF1R* promoter stimulation.

### 3.8. Binding of ERK1/2 and AKT to the Proximal IGF1R Promoter

To investigate the capacity of both ERK1/2 and AKT to physically bind to the *IGF1R* promoter region, DNA affinity chromatography assays were conducted using a 511-bp proximal fragment of the *IGF1R* sequence. To this end, a biotinylated DNA fragment extending from position -458 to +53 (encompassing the *IGF1R* gene transcription initiation site) was bound to streptavidin magnetic beads and incubated with nuclear extracts of IGF1R-KD, INSR-KD or control MCF7 cells. Bound proteins were eluted with a high salt buffer and analyzed by Western blots. Results obtained provide evidence that both ERK1/2 and AKT proteins bind to the *IGF1R* promoter region (Figure 8A,B). Binding intensity, however, was reduced in INSR-KD cells. On the other hand, phospho-ERK1/2 seems to bind to the *IGF1R* promoter only in the INSR-KD cell line. Neither protein was detected in association with the promoter in IGF1R-KD cells. 

### 3.9. Effect of IGF1R/INSR Silencing on Cell Proliferation

To assess the impact of IGF1R or INSR silencing on MCF7 and T47D cell proliferation, cells were seeded in 96-well plates at a density of 6 × 10^3^ (MCF7) or 1 × 10^4^ (T47D) cells/well. After 24 h, the medium was replaced with starvation media in the presence or absence of IGF1 or insulin (50 ng/mL) for an additional 72 h. Results of XTT assays on MCF7 cells indicate that IGF1R-KD cells exhibited an enhanced proliferation, suggesting that the INSR signaling pathway that operates in these cells drives a stronger proliferative response (Figure 9A). Consistent with these results, insulin stimulated proliferation of control cells by 200% compared to 145% stimulation by IGF1. In addition, no change in basal proliferation rate was seen in INSR-KD, compared to control, cells. As expected, insulin, but not IGF1, stimulated proliferation of IGF1R-KD cells, whereas IGF1, but not insulin, stimulated proliferation of INSR-KD cells. Analysis of cell proliferation in T47D-derived cells indicates that IGF1 treatment of INSR siRNA-transfected cells led to a 38% increase in proliferation compared to untreated cells, whereas insulin treatment resulted in a 48% reduction in proliferation (Figure 9B). Unlike MCF7 cells, neither IGF1 nor insulin had any effect on IGF1R-KD T47D cells.

### 3.10. Effect of IGF1R/INSR Silencing on Cell Migration

Finally, to address the effect of IGF1R/INSR silencing on MCF7 cell migration, scratch assays were conducted. Scratches were done using WoundMaker, after which cells were kept in starvation media in the presence or absence of IGF1 or insulin for 96 h. Analysis of cell migration distances revealed that wound closure was seen at 96 h in control cells treated with IGF1 or insulin (Figure 10A,B). IGF1 induced an increase in migration in INSR-KD cells (60%), whereas insulin stimulated wound closure in IGF1R-KD cells by 43%, suggesting that each hormone was working through its specific receptor. Finally, ligand-induced migration was inhibited in cells with a silenced IGF1R or INSR in comparison with the respective untreated cells. The impact of cell proliferation on wound closure cannot be discarded.

## 4. Discussion

The IGF1R plays a critical role in establishment and maintenance of the malignant phenotype [30,31,32,33]. While the insulin signaling pathway has been classically associated with metabolic types of action, evidence accumulated over the past several years identified the INSR as an important player in cancer development. The interplay between INSR and IGF1R and their downstream cytoplasmic mediators is a topic of major basic and clinical interest [2,5,34,35]. Thus, recent studies demonstrate that INSR isoform-A modulates metabolic reprogramming of breast cancer cells in response to both IGF2 and indulin stimulation [36].

Recent morphological, biochemical and molecular analyses conducted by a number of laboratories led to the unequivocal confirmation that INSR and IGF1R translocate to cell nucleus in different types of cells and tissues, including normal non-malignant cells [10,11,12,13]. While receptor sumoylation appears to play an important role in the internalization process, evidence has been provided showing that this post-translational modification is not a vital prerequisite [11]. Furthermore, the question of whether the subcellular distribution of the INSR or IGF1R might be selectively affected by the specific ligand stimulating the receptor is yet to be experimentally addressed. Nuclear INSR and IGF1R, either directly or as part of multi-protein complexes, bind to a number of gene promoters and enhancer elements in, apparently, a sequence-specific manner. Hence, these nuclear receptors fall within the functional definition of transcription factors [17]. Whilst the genomic roles of INSR/IGF1R have not yet been fully elucidated, it is clear that this global regulatory mechanism provides a further level of biological complexity. An additional level of adaptation is suggested by the recent finding that hybrid receptors, composed of an INSR hemireceptor linked to an IGF1R hemireceptor, are capable of translocating to nucleus in epithelial cells [37]. The endocytotic pathway involved in the endosomal sorting of membrane receptors has been described in detail [38].

Mitogens, including insulin and IGF1, induce a rapid (5–10 min) activation of ERK1/2, which, under resting conditions, are anchored in the cytoplasm [39]. This increase in kinase activity is followed by a second wave that continues for several hours. Nuclear translocation of ERK1/2 takes place immediately following external stimuli and remains during the entire G1 phase [40]. It has been shown that nuclear transport of ERK1/2 is of critical importance for the cell to progress from G1 to S phases [41]. AKT constitutes a key mediator of multiple growth factor- and G-protein-stimulated cellular processes. Three AKT-encoding genes exist in mammals, displaying a high degree of homology in their catalytic domains [42]. Following activation, AKT isoforms are translocated to nucleus, where they interact and phosphorylate a number of substrates [29].

The availability of MCF7 breast cancer cells with specific abrogation of the INSR or IGF1R allowed us to examine the effects of these pathways on the subcellular distribution and activation of the receptors as well as the downstream effectors. The existence of a compensatory mechanism responsible for enhanced IGF1R levels following INSR silencing was suggested by experiments showing that high IGF1R expression was seen in INSR-KD cells. Of interest, IGF1R, but not INSR, silencing had a significant effect on nuclear ERK1/2 activation. These results are consistent with the notion that the IGF1 signaling pathway has a leading role in nuclear ERK1/2 activation. Of interest, co-IP assays indicate that nuclear ERK1/2 and AKT were sumoylated, suggesting that, similarly to the nuclear INSR and IGF1R [12], this post-translational modification might be physiologically important in the context of cytoplasmic mediators translocation. Experiments performed in MCF7 cells were corroborated in the T47D cell line, although sometimes with different trends. The rationale for this choice of cells is the fact that both cell lines are molecularly defined as Luminal A (ER+/PR+/HER2-). SiRNAs were employed to specifically abrogate IGF1R or INSR expression in this cell line. Similarly to MCF7 cells, a decrease in nuclear AKT, but not ERK1/2, was seen in IGF1R-KD T47D cells. Of interest, T47D cells display strong phosphorylation of the high MW (~250-kDa) pro-receptors. While the molecular basis for this atypical phenotype is unclear, the fact that T47D cells express a mutant *p53* gene (unlike MCF7 cells that contain a wild-type *p53*) may partly explain some of the differences between cells.

Transfection assays revealed that basal *IGF1R* promoter activity was reduced by 27% in IGF1R-KD cells and by 54% in INSR-KD cells. These results indicate that both IGF1 and insulin pathways are needed to achieve maximal promoter activity, although the insulin path has a preponderant effect. However, we must take into consideration that there is a large overlap between these pathways and, therefore, these analyses are difficult to quantitate and interpret. Consistent with the lack of a functional IGF1R, IGF1 was unable to stimulate *IGF1R* promoter activity in IGF1R-KD cells. IGF1, however, had a strong stimulatory effect on INSR-KD cells.

Of interest, we demonstrate that both ERK2 and AKT are capable of stimulating *IGF1R* promoter activity, although the intensity of this effect was diminished in cells with disrupted IGF1R or INSR. These data are consistent with the concept that, in addition to their prototypical role as targets for insulin and IGF1 action, both ERK and AKT can control transcription of the *IGF1R* gene. Moreover, results provide evidence of complex interactions between ERK and AKT, and IGF1R protein in control of the *IGF1R* promoter. Thus, co-transfection of an IGF1R, but not an INSR, expression construct diminished the ability of ERK2 to stimulate *IGF1R* promoter activity. These functional interactions were confirmed by co-IP assays that revealed that IGF1R and ERK1/2 exhibited a stronger physical interaction in INSR-KD than in IGF1R-KD cells. Hence, ERK1/2 binds IGF1R with a larger affinity than it binds INSR.

In addition to the protein–protein interactions described above, DNA affinity chromatography assays revealed that both ERK1/2 and AKT proteins bind to the *IGF1R* promoter region. Binding intensity, however, was reduced in INSR-KD cells. Of interest, phosphorylated ERK1/2 binds to the *IGF1R* promoter in INSR-KD, but not IGF1R-KD, cells. The biochemical and cellular basis for this differential binding is still unclear. Nuclear AKT and ERK1/2 are involved in key signaling pathways, including cell cycle progression, DNA repair and RNA export. While it is unknown whether ERK1/2 and AKT bind directly to DNA or as part of multi-protein complexes, these novel functional and physical interactions might help understand the mechanisms that modulate insulin signaling pathways in the nucleus.

In terms of the biological activities evaluated here, we observed that MCF7 cells with a silenced IGF1R displayed an augmented mitogenic response. These results suggest that the insulin signaling pathway drives a more robust proliferative reaction. Proliferation assays were complemented by scratch assays that showed that IGF1 stimulated migration in INSR-KD cells by 60%, whereas insulin enhanced wound closure in IGF1R-KD cells by 43%. Ligand-induced migration was inhibited in cells with silenced IGF1R or INSR, indicating that both receptors are involved in breast cancer cell migration.

## 5. Conclusions

The current results demonstrate that both INSR and IGF1R exhibit a nuclear localization in breast cancer-derived cells. In addition, the insulin and IGF1 pathways have different effects on the subcellular distribution (and, particularly, the nuclear presence) of ERK1/2 and AKT molecules. Both cytoplasmic mediators are capable of binding and transactivating the *IGF1R* promoter. These data are consistent with the notion that, in addition to their classical role as targets for insulin-like molecules, following nuclear translocation both ERK1/2 and AKT are involved in transcriptional control of the *IGF1R* gene. The physiological and clinical implications of this feed-back loop need to be further investigated. Given the emerging role of INSR and IGF1R as potential therapeutic targets [43], it will be of clinical importance to assess the potential use of nuclear receptors and mediators as biomarkers for INSR/IGF1R targeted therapy.

## Figures and Tables

**Figure 1 cells-08-01499-f001:**
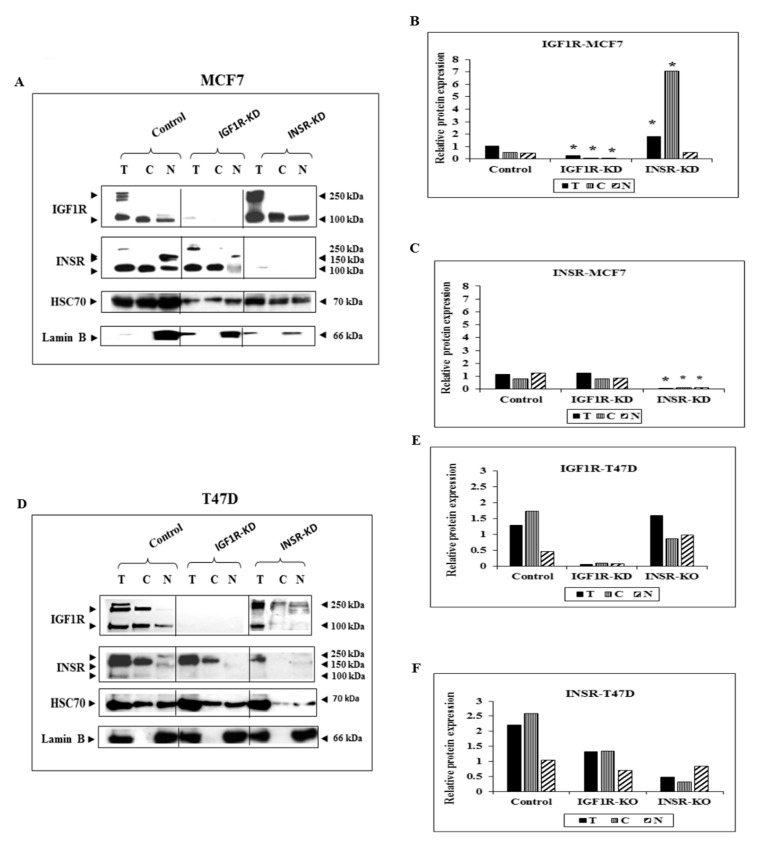
Subcellular analysis of IGF1R and INSR expression in MCF7-derived IGF1R KD and INSR KD cells and IGF1R/INSR siRNA-transfected T47D cells. (**A**) MCF7 cells were transfected with shRNAs against IGF1R or INSR (or control shRNA), selected for puromycin resistance, and fractionated into cytoplasmic and nuclear fractions. Western blot analysis of total IGF1R and INSR was conducted using antibodies against total IGF1R and INSR. Lanes with total (T) extracts include 100 μg protein and lanes with cytoplasmic (C) or nuclear (N) extracts include 40 μg protein. Heat shock cognate protein-70 (HSC70) was used as a marker for total protein and lamin B was used as a marker for nuclear fractions. (**B**,**C**) Scanning densitometry analysis of basal IGF1R and INSR levels. Bars represent IGF1R and INSR values (AU, arbitrary units), normalized to the corresponding HSC70 levels. Results of an illustrative experiment, repeated twice with similar results, are shown. *, *p* < 0.01 versus control cells. (**D**) T47D cells were transfected with siRNA against IGF1R and INSR, or negative control siRNA, and after 72 h IGF1R and INSR were detected by Western blots. (**E**,**F**) Optical density is expressed as IGF1R or INSR values normalized to the corresponding HSC70.

**Figure 2 cells-08-01499-f002:**
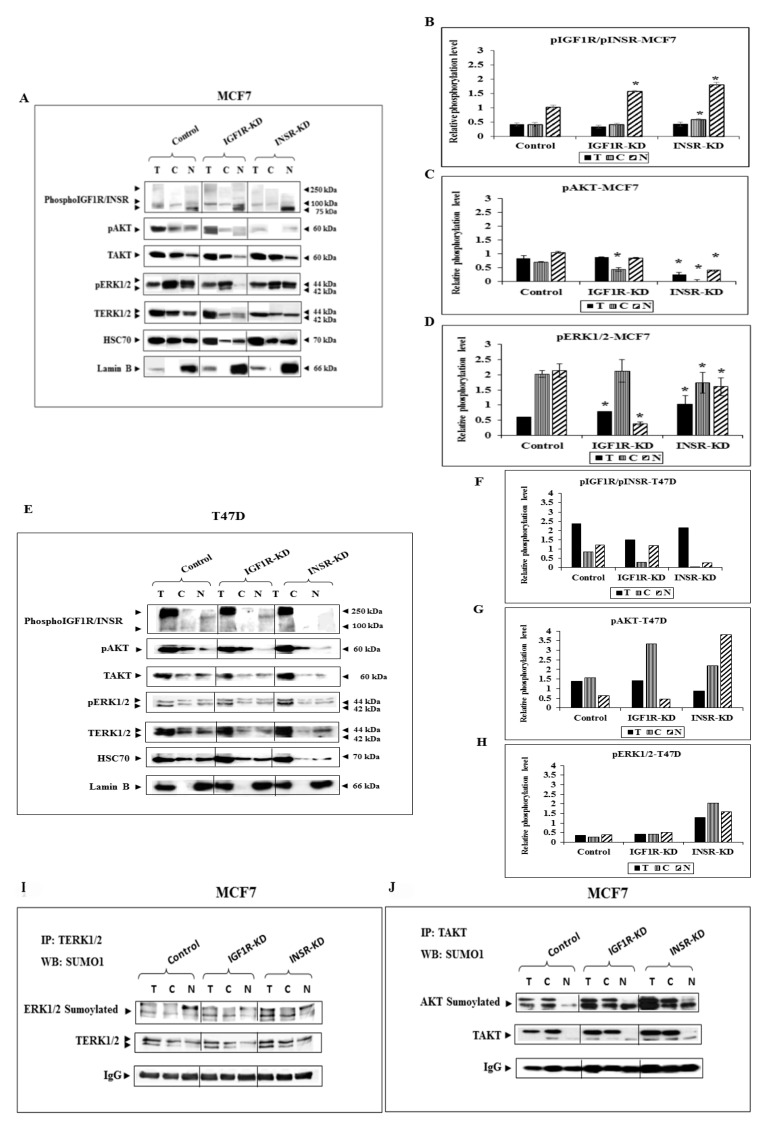
Western blot analysis of ERK1/2 and AKT expression in IGF1R KD and INSR KD cells. (**A**) MCF7-derived IGF1R-KD, INSR-KD and control cells were fractionated into cytoplasmic and nuclear fractions, and Western blots were conducted using antibodies against phospho-IGF1R/INSR and total and phospho ERK1/2 and AKT. Lanes with total (T) cell extracts include 100 μg protein and lanes with cytoplasmic (C) or nuclear (N) extracts include 40 μg protein. HSC70 was used as a loading control and lamin B was used as a nuclear marker. (**B**–**D**) The intensities of the bands were quantified and relative phosphorylation levels were calculated by correcting phosphorylation levels to HSC70, total AKT or ERK1/2 levels. Data represents three independent experiments (mean ± SEM; *n* = 3; *, *p* < 0.01 versus control cells). (**E**) Detection of phospho-IGF1R/INSR, pAKT and phopho-ERK1/2 in T47D cells transfected with siRNAs against IGF1R, INSR, or non-targetting (NT). (**F**–**H**) Optical density is expressed as phospho-IGF1R/INSR, phospho-AKT or phospho-ERK1/2 levels normalized to the corresponding HSC70, total AKT or ERK1/2 levels. (**I**, **J**) Total cell extracts (500 µg), and cytosolic (200 µg) and nuclear (200 µg) fractions of control, IGF1R-KD and INSR-KD cells MCF7 cells were immunoprecipitated with anti-total ERK1 combined with anti-total ERK2 or anti-total AKT, followed by SDS-PAGE and immunoblotting with TERK1/2, TAKT or Sumo1 antibodies. IgG was used as a control for the co-IP experiments.

**Figure 3 cells-08-01499-f003:**
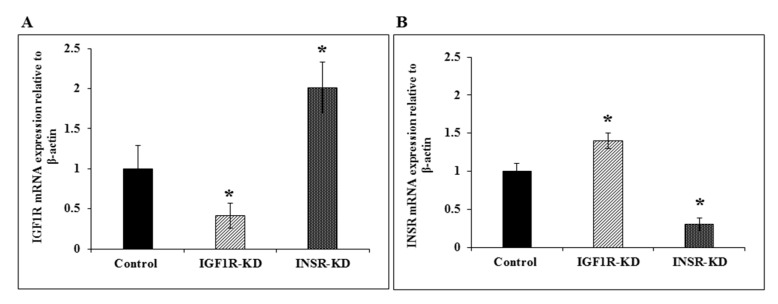
Real time-quantitative PCR analysis of IGF1R and INSR mRNA levels. Confluent MCF7-derived IGF1R-KD, INSR-KD and control cells were harvested and total RNA was prepared, as described in Section 2. Levels of IGF1R (**A**) and INSR (**B**) mRNAs were measured by RT-QPCR using primers described in Table 1. For each mRNA, a value of 1 was given to the level displayed by controls. Bars denote mean ± SEM (*n* = 3). *, *p* < 0.05 versus respective control.

**Figure 4 cells-08-01499-f004:**
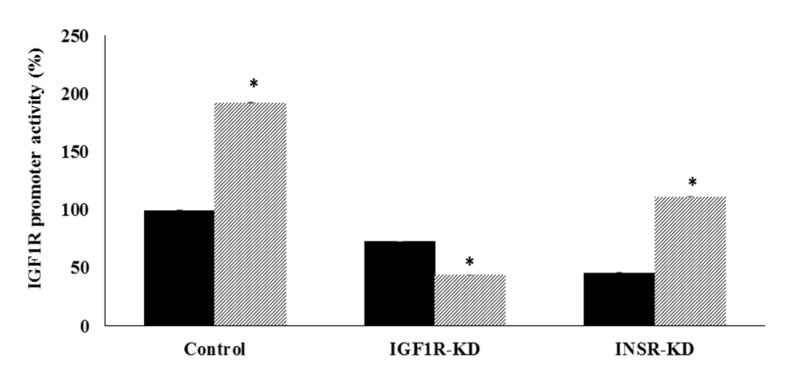
Effect of IGF1 on *IGF1R* promoter activity in IGF1R KD and INSR KD cells. MCF7-derived IGF1R KD, INSR KD and control cells were transiently transfected with a proximal *IGF1R* promoter luciferase reporter construct [p(−476/+640)LUC]. After 24 h, cells were treated with IGF1 (50 ng/mL) (dotted bars), or left untreated, for control purposes (solid bars) for an additional 24 h, after which cells were harvested and luciferase levels were measured. Promoter activities are expressed as luciferase values normalized to total protein. A value of 100% was given to the promoter activity generated by the reporter plasmid in control MCF7 cells. Bars represent mean ± SEM of three independent experiments in duplicate wells. *, *p* < 0.05 versus control cells.

**Figure 5 cells-08-01499-f005:**
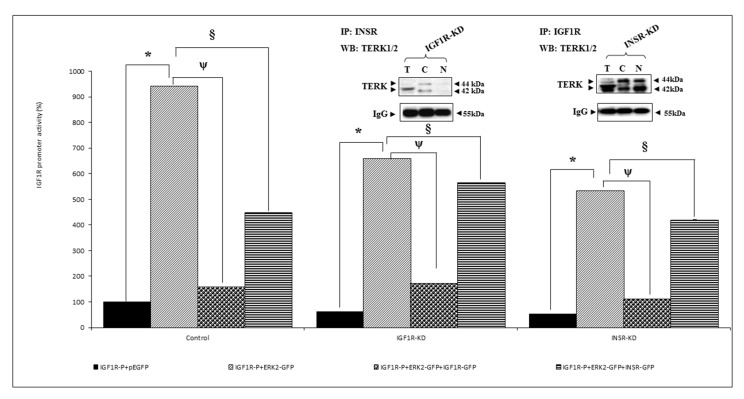
Effect of ERK2 on *IGF1R* promoter activity. IGF1R KD, INSR KD and control MCF7 cells were co-transfected with the p(−476/+640)LUC *IGF1R* promoter, along with an ERK2-GFP expression vector (dotted bars), or empty GFP vector (solid bars). In addition, transfection mix included an IGF1R-GFP expression vector (circles-filled bars) or an INSR-GFP expression vector (horizontally hatched bars). After 48 h, cells were harvested and luciferase activity was measured. Promoter activities are expressed as luciferase values normalized to total protein. A value of 100% was given to the promoter activity generated by the reporter plasmid in control MCF7 cells. *, *p* < 0.01 ERK2-co-transfected cells versus empty vector co-transfected cells; ^ψ^, *p* < 0.01 ERK2 + IGF1R-GFP-co-transfected cells versus empty vector co-transfected cells; ^§^, *p* < 0.01 ERK2 + INSR-GFP-co-transfected cells versus empty vector co-transfected cells. Total, cytoplasmic and nuclear fractions of IGF1R-KD and INSR-KD cells were immunoprecipitated with anti-INSR or anti-IGF1R, respectively, and immunoblotted with an ERK1/2 antibody, as described in Section 2 (insets). The 42- and 44-kDa total ERK bands are indicated. IgG was used as a control for the co-IP experiment. All experiments were conducted in triplicates.

**Figure 6 cells-08-01499-f006:**
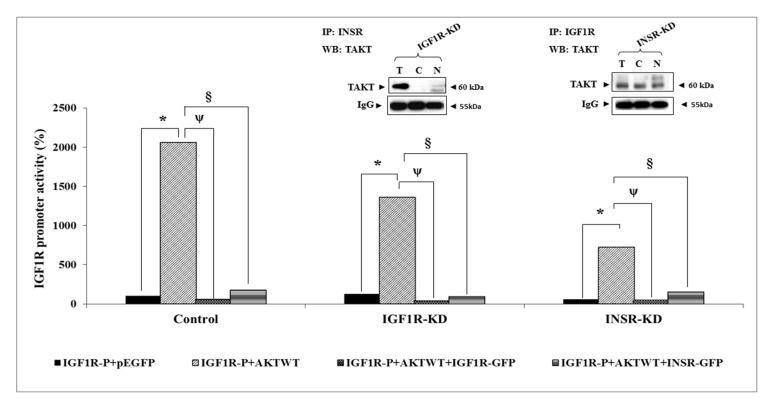
Effect of AKT on *IGF1R* promoter activity. IGF1R-KD, INSR-KD and control MCF7 cells were co-transfected with the p(−476/+640)LUC *IGF1R* promoter along with a wild-type AKT expression vector (dotted bars), or empty GFP vector (solid bars). In addition, transfection mix included an IGF1R-GFP expression vector (circles-filled bars) or an INSR-GFP expression vector (horizontally hatched bars). After 48 h, cells were harvested and luciferase activity was measured. Promoter activities are expressed as luciferase values normalized to total protein. A value of 100% was given to the promoter activity generated by the reporter plasmid in control MCF7 cells. *, *p* < 0.01 AKTWT-co-transfected cells versus empty vector co-transfected cells; ^ψ^, *p* < 0.01 AKTWT + IGF1R-GFP-co-transfected cells versus empty vector co-transfected cells; ^§^, *p* < 0.01 AKTWT + INSR-GFP-co-transfected cells versus empty vector co-transfected cells. Total, cytoplasmic and nuclear fractions of IGF1R-KD and INSR-KD cells were immunoprecipitated with anti-INSR or anti-IGF1R, respectively, and immunoblotted with an AKT antibody (insets). The 60-kDa total AKT band is indicated. IgG was used as a control for the co-IP experiment. All experiments were conducted in triplicates.

**Figure 7 cells-08-01499-f007:**
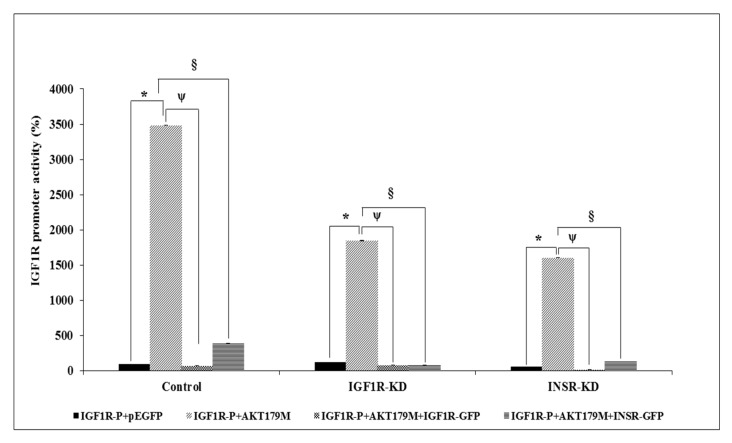
Effect of mutant AKT on *IGF1R* promoter activity. Co-transfection experiments were conducted as described in the Legend to Figure 6, using a mutant AKT expression vector (K179M). A value of 100% was given to the promoter activity generated by the reporter plasmid in control MCF7 cells. *, *p* < 0.01 AKT179M-co-transfected cells versus empty vector co-transfected cells; ^ψ^, *p* < 0.01 AKT179M + IGF1R-GFP-co-transfected cells versus empty vector co-transfected cells; ^§^, *p* < 0.01 AKT179M + INSR-GFP-co-transfected cells versus empty vector co-transfected cells.

**Figure 8 cells-08-01499-f008:**
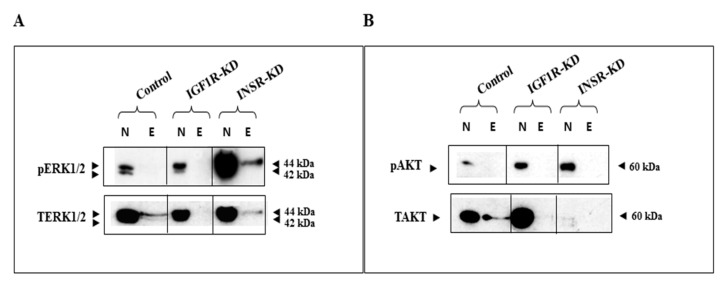
Binding of ERK1/2 and AKT to *IGF1R* promoter DNA. DNA affinity chromatography was conducted by labeling a fragment (−458 to +53) of the *IGF1R* gene promoter with biotin. The labeled fragment was bound to streptavidin magnetic beads and incubated with nuclear (N) extracts of IGF1R-KD, INSR-KD or control cells. Bound proteins were eluted (E) with high salt buffer and analyzed by Western blots using ERK1/2 (**A**) and AKT (**B**) antibodies.

**Figure 9 cells-08-01499-f009:**
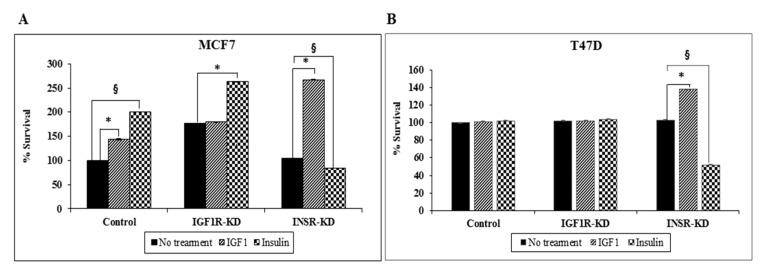
Effect of IGF1R/INSR silencing on cell proliferation. IGF1R-KD, INSR-KD and control MCF7 cells (**A**) and IGF1R siRNA-, INSR siRNA- or control siRNA-transfected T47D cells (**B**) were plated in 96-well plates. After 24 h, the medium was replaced with starvation media in the absence (solid bars) or presence of IGF1 (cross-hatched bars) or insulin (squares-filled bars) (50 ng/mL) for an additional 72 h (MCF7) or 48 h (T47D), after which cell viability was determined by an XTT assay. A value of 100% was given to the cell number of control, untreated cells at the end of the incubation period. *, *p* < 0.05 IGF1-treated *versus* control cells; ^§^, *p* < 0.05 insulin-treated versus control cells. All experiments were conducted in triplicates.

**Figure 10 cells-08-01499-f010:**
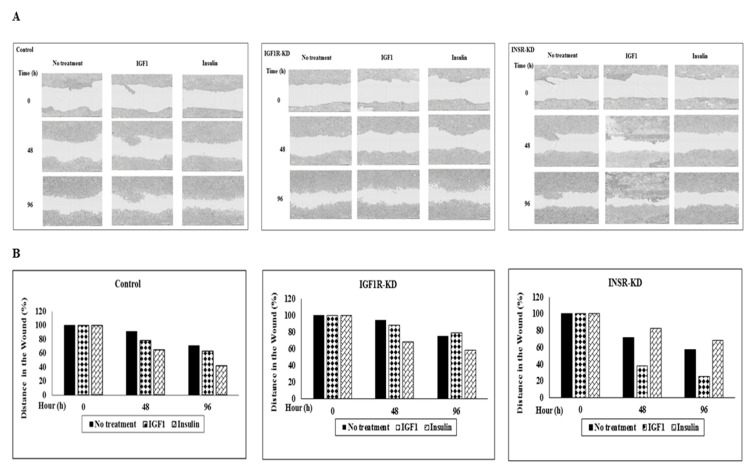
Effect of IGF1R/INSR silencing on cell migration. (**A**) A scratch was done using WoundMaker in IGF1R-KD, INSR-KD and control MCF7 cells. The cells were then treated with starvation media in the presence or absence of IGF1 or insulin for 96 h. The plates were placed into an IncuCyte ZOOM system, with repeat scanning every 2 h. Representative micrographs taken at start, 48 h and 96 h are shown. Results were in quadriplicates and every wound was photographed in two different areas. (**B**) Scanning densitometry analysis of cell migration distances in control, IGF1R-KD and INSR-KD cells, treated with IGF1 or insulin for 48 or 96 h. A value of 100% was given to the distance in the wound at time 0. Bars correspond to the micrographs depicted in panel A.

**Table 1 cells-08-01499-t001:** Sequences of primers used in RT-QPCR assays.

Primer	Sequence	Product Size
IGF1R-F	5′-GTGGAGACAGGGGCTTTTATT-3′	122 bp
IGF1R-R	5′-CTCCAGCCTCCTTAGATCACA-3′	
INSR-F	5′-AGTGTGGAGACATCTGTCCG-3′	121 bp
INSR-R	5′-GTCGGGCAAACTTTCTGGCA-3′	
βActin-F	5′-CCTGGCACCCAGCACAAT-3′	144 bp
βActin-R	5′-GGGCCGGACTCGTCATACT-3′

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
