# Peer review of "Differential Effects of Insulin and IGF1 Receptors on ERK and AKT Subcellular Distribution in Breast Cancer Cells"

_cells, 2019, doi:10.3390/cells8121499_

Round 1
Reviewer 1 Report
This is a very important study investigating the subcellular distribution of various components of the IGF1R/IR signaling pathways. The paper is well-written and easy to follow, the experimental approach is sound and the conclusions fully supported by data. The only suggestion I have is to include in the discussion section the hypothesis that subcellular distribution of the IGF1R/IR or other signaling components might be selectively affected by the type of the ligand stimulating the receptor. This might be especially important for the IGF1/IGF2/insulin as well as for other non-canonical agonists of the IIGFR family.
Author Response
We thank Reviewer 1 for his/her comments. A statement regarding the potential differential effects of the various ligands activating the receptors on the subcellular distribution of the INSR and IGF1R has been incorporated in the Discussion (lines 440-442).
Reviewer 2 Report
The paper investigates similarities and disparities between the closely related IR and IGF-1R signalling systems. The differential subcellular localization of downstream components that the authors describe may in part explain why distinct functional outcomes arise from largely similar signal components.
General comments: Texts in some figures are too small, and cannot be made out when manuscript is printed out on A4. The outlines around figure subsections are also not needed. Authors should consider combining Figures 5,6,7 in one figure.
Specific comments:
In the subcellular fractionation experiments, is there a cytoplasm control that can be used (e.g. Tubulin?), as currently the possibility of nuclear contamination in the cytoplasmic fraction cannot be ruled out.
Numerous graphs lack error bars, therefore I cannot assess the spread of the data between experimental replicates.
The scratch assay runs over a period of 96 h - within this time window the cells are also proliferating under stimulated conditions (as illustrated in figure 9). The scratch assay starts with a high confluency monolayer of cells and therefore over this time window, wound closure is likely also contributed by the proliferation of cells, not solely migration. Such a scenario should be at the least acknowledged, or more stringent conditions carried out for migration to be assessed, e.g. time window below the cell doubling time, or transwell-style assays.
Author Response
General comments:
Texts in some figures are too small, and cannot be made out when manuscript is printed out on A4
Figures were now enlarged in order to better visualize the text inside the figures.
Authors should consider combining Figures 5,6,7 in one figure.
Figures 5, 6 and 7 are now presented as Figure 5A, B and C.
Specific comments:
In the subcellular fractionation experiments, is there a cytoplasm control that can be used (e.g. Tubulin?).
Tubulin cannot be used as a cytoplasmic marker because it has been reported to be present in nucleus (Gozes et al, JBC 252:1819, 1977). In a more general sense, it is difficult to find a cytoplamic protein that is never present in nucleus.
Numerous graphs lack error bars, therefore I cannot assess the spread of the data between experimental replicates.
When not indicated, SEM bars are smaller than the symbol size.
The scratch assay runs over a period of 96 h - within this time window the cells are also proliferating under stimulated conditions (as illustrated in figure 9). The scratch assay starts with a high confluency monolayer of cells and therefore over this time window, wound closure is likely also contributed by the proliferation of cells, not solely migration. Such a scenario should be at the least acknowledged, or more stringent conditions carried out for migration to be assessed, e.g. time window below the cell doubling time, or transwell-style assays.
We thank Reviewer 2 for this comment. A statement has been incorporated indicating that wound closure can be also affected by the proliferation of the cells (lines 416-417).
Reviewer 3 Report
In this manuscript, the authors show that insulin and IGF1 pathways have differential effects on subcellular distribution and activation of ERK1/2 and AKT. They also showed that both ERK1/2 and AKT are capable of stimulating IGF1R promoter activity.
This study is well designed. Nevertheless, for a better understanding, some points need to be cleared up before this manuscript may be considered for publication.
Major points
The figures are difficult to read, especially since the characters are too small. Why have authors shown their data as mean+/-SEM (fig 4), mean +/-SD (fig 2 B C D; fig 3), or most often have not indicated (fig 1 E F, fig 2 EFG, fig 5, fig 6, fig 7, fig 9, fig 10). The number of experiments is also forgotten in most legends of the figures. Sometimes the experiments were repeated 2 times, 3 times, or? Why? It is therefore difficult to judge the quality of the work. *, **, *** in the legend of fig 6 are not present in the graph. Contrary to what is stated the legend of figure 1, data with phosphorylated IGF1R and INSR are shown in fig 2. The authors have to read more carefully the manuscript and to complete the legends of the figures. Some experiments were not carried out on T47D. Why? In the insert of fig 5 IP INSR(left panel) in the total extract you show one band of tERK at 43kd but two bands (42 and 44 kd) in the cytosol. How do you explain this? Lines 410-412: Unlike it is stated in this sentence I don’t see “a complete wound closure” in any condition in figure 10. This sentence should be re-writen. Could you explain the two sentences lines 412-414? You say “IGF1 induced a marked increase in migration in INSR KD cells…” and in the next sentence “Finally, ligand-induced migration was inhibited in cells with a silenced IGF1R or INSR.”Minor points
Lines 200-202: this sentence seems to be related to figure 1B. If true this figure should be cited at the end of the sentence. The abbreviations TERK and TAKT are not explained. You should use tERK and tAKT instead.Author Response
The figures are difficult to read, especially since the characters are too small.
As indicated above, figures were enlarged.
Why have authors shown their data as mean+/-SEM (fig 4), mean +/-SD (fig 2 B C D; fig 3), or most often have not indicated (fig 1 E F, fig 2 EFG, fig 5, fig 6, fig 7, fig 9, fig 10).
We apologize for this oversight. Only SEM is presented now.
The number of experiments is also forgotten in most legends of the figures. Sometimes the experiments were repeated 2 times, 3 times, or? Why?
The number of experiments is now clearly stated in all legends to figures. In the few events in which only two replicates are presented, it is because results were very similar.
It is therefore difficult to judge the quality of the work. *, **, *** in the legend of fig 6 are not present in the graph.
Asterisks were removed and we present different symbols to denote statistical differences.
Contrary to what is stated the legend of figure 1, data with phosphorylated IGF1R and INSR are shown in fig 2. The authors have to read more carefully the manuscript and to complete the legends of the figures.
We apologize for this mistake. Reference to phosphorylated proteins was deleted.
Some experiments were not carried out on T47D. Why?
Characterization studies were conducted on both cell lines. However, more complex biological and promoter analyses were conducted only in MCF7 cells.
In the insert of fig 5 IP INSR(left panel) in the total extract you show one band of tERK at 43kd but two bands (42 and 44 kd) in the cytosol. How do you explain this?
Careful examination of the inset reveals a weak band at 44 kDa in the total extract. A similar pattern is seen in the right panel. It is difficult to explain the differences between the total extract and subcellular fractions.These differences, however, are commonly seen.
Unlike it is stated in this sentence I don’t see “a complete wound closure” in any condition in figure 10. This sentence should be re-writen. Could you explain the two sentences lines 412-414? You say “IGF1 induced a marked increase in migration in INSR KD cells…” and in the next sentence “Finally, ligand-induced migration was inhibited in cells with a silenced IGF1R or INSR.”
These sentences were now entirely rephrased and we believe that they are clear now.
Minor points
Lines 200-202: this sentence seems to be related to figure 1B. If true this figure should be cited at the end of the sentence.
This mistake was corrected.
The abbreviations TERK and TAKT are not explained. You should use tERK and tAKT instead
The abbreviations TERK and TAKT are commonly used in the literature to denote Total ERK and Total AKT.
Round 2
Reviewer 3 Report
This manuscriupt is now publishable.
Author Response
All three references were now included in the Discussion